# Time-Domain Aeroelasticity Analysis by a Tightly Coupled Fluid-Structure Interaction Methodology

**Zhongyu Liu, Xueyuan Nie \*, Guannan Zheng \* and Guowei Yang \***

Key Laboratory for Mechanics in Fluid Solid Coupling Systems, Institute of Mechanics, Chinese Academy of Sciences, Beijing 100190, China; zhongyu.liu@cicc.com.cn
\* Correspondence: niexueyuan@imech.ac.cn (X.N.); zhengguannan@imech.ac.cn (G.Z.); gwyang@imech.ac.cn (G.Y.)

**Abstract:** A tightly coupled fluid-structure interaction (FSI) methodology is developed for aeroelasticity analysis in the time domain. The preconditioned Navier–Stokes equations for all Mach numbers are employed and the structural equations are tightly coupled with the fluid equations by discretizing their time derivative term in the same pseudo time-stepping method. A modified mesh deformation method based on reduced control points radial basis functions (RBF) is utilized, and a RBF based mapping algorithm is introduced for data exchange on the interaction interface. To evaluate the methodology, the flutter boundary and the limit cycle oscillation of Isogai wing and the flutter boundary of AGARD 445.6 wing are analyzed and validated.

**Keywords:** CFD/CSD coupling; flutter boundary; computational aeroelasticity





## 1. Introduction

Aeroelasticity phenomenon, which arises from the interaction of the aerodynamic, elastic and inertial forces, may have detrimental effect on the reliability, cost and safety of aircraft. However, the complicated interaction mechanism and multidisciplinary requirement of aeroelasticity problem make it difficult to be analyzed with efficiency and accuracy.

With the development of high performance computers, coupled fluid-structure interaction (FSI) methodology is made available for aeroelasticity precondition and efforts have been made to develop the precondition capability of these methods. Generally, coupled FSI methodology can be divided into three types: the fluid and structural models are fully coupled, loosely coupled or tightly coupled. In the fully coupled method, the fluid and structural equations are integrated in both Eulerian and Lagrangian systems and solved in time simultaneously. The fully coupled method has a requirement in limitations on grid size. Moreover, this leads to the matrices being orders of magnitudes stiffer for structure system than fluid system, which makes it virtually impossible to solve the equations for large-scale problems [1]. For the loosely and tightly coupled methods, they use the existing solvers for computational fluid dynamics (CFD) and computational structure dynamics (CSD) [2,3]. Because the aerodynamic and structure domains are discretized in a different manner, data communications at the fluid–structure interface are required. The difference between the loosely and tightly coupled methods in essence lies in the data communication times. The loosely coupled strategy transfers the aerodynamic forces and structural displacements between fluid and structure systems only once in a physical time step, so it is usually only first-order time accuracy. For tightly coupled methods, data communications are performed several times within each physical time step and second-order time accuracy can be obtained. The linear or nonlinear types of models can be coupled together in FSI methodology. Current industrial practice relies heavily on linear methods, which can lead to conservative design and flight envelope restrictions [4]. However, FSI methodology with linear aerodynamic model cannot reveal the nonlinear aeroelastic effects, such as the transonic flutter dip and the limit cycle oscillations.

During the analysis of aeroelasticity problem, flexible structure deforms under the influence of air-loads, so it's necessary to update the fluid volume mesh according to the structural displacement. In the past decades, several mesh deformation methods have been investigated and successfully applied to aeroelastic analysis. Recently, extensive attention has been paid to the mesh deformation method based on radial basis functions (RBF), which can provide high-quality mesh deformation and perform reliable capability for arbitrary topology. The efficiency of RBF method is related to the number of control points on the wall surfaces of fluid mesh, and a reduced control points method proposed by Rendall [5] greatly improves the efficiency, which makes it possible to deal with large meshes. Unfortunately, the gain in deformation efficiency may cause the problem of deformation error on deformed surfaces and inverted boundary layer meshes near wall.

Computational fluid dynamics (CFD) and computational structural dynamics (CSD) based FSI methodology become attractive for aeroelasticity analysis in the time domain. As CFD and computer technology progress, higher-order methods based on RANS coupled with CSD are preferable because they are able to model more accurately transonic, nonlinear, and viscous effects, compared with linear aerodynamic models. However, the traditional compressible RANS solver may lead to low precision and low efficiency solutions at low Mach number, on account of the overlarge numerical dissipation and condition number [6]. To overcome these problems, an effective method is to use the preconditioned method with multiplying the preconditioning matrix to the time-derivative term, which can reduce the disparity between the convective and acoustic wave velocities through the modification of the eigensystem of the governing equations. The preconditioned method has already been applied to the flow analysis [7–9], but it hasn't been coupled with CSD for FSI application yet. In the current study, a complete integrated CFD-CSD computational method is implemented for aeroelasticity analysis in the time domain. CFD solver adopts the preconditioned Navier–Stokes governing equations, which can provide accurate flow solutions for both compressible and incompressible flows, are solved to evaluate aerodynamic loads, especially the nonlinear loads in the transonic regime. The structural model is tightly coupled with the fluid model by a pseudo time-stepping temporal discretization method. A modified mesh deformation method based on reduced control points RBF is utilized to update the fluid volume mesh, and an RBF based mapping algorithm is employed for data exchange on the fluid-structure interface. To evaluate the validity of the developed methodology, the flutter boundary and the limit cycle oscillation of Isogai wing and the flutter boundary of AGARD 445.6 wing, both of which have typical nonlinear characteristics, are analyzed and validated by experimental data.

## 2. Fluid Model

### 2.1. Preconditioned Governing Equations

The unsteady preconditioned Navier–Stokes governing equations are adopted to predict the aerodynamic loads during the analysis of aeroelasticity from incompressible to compressible flows [6]:

$$\mathbf{\Gamma}\frac{\partial \hat{W}}{\partial \tau} + \frac{\partial \hat{Q}}{\partial t} + \frac{\partial \hat{E}}{\partial \xi} + \frac{\partial \hat{F}}{\partial \eta} + \frac{\partial \hat{G}}{\partial \zeta} = \frac{\partial \hat{E}_v}{\partial \xi} + \frac{\partial \hat{F}_v}{\partial \eta} + \frac{\partial \hat{G}_v}{\partial \zeta} \tag{1}$$

where $\hat{W} = (p\ u\ v\ w\ T)^T/J$ is the primitive variable vector $\hat{Q} = \begin{pmatrix} \rho & \rho u & \rho v & \rho w & \rho e \end{pmatrix}^T/J$ is the conservative variable vector. The preconditoner $\mathbf{\Gamma}$ has various forms and the type proposed by Weiss–Smith is introduced here:

$$\mathbf{\Gamma} = \begin{pmatrix} \Theta & 0 & 0 & 0 & \rho_T \\ \Theta u & \rho & 0 & 0 & \rho_T u \\ \Theta v & 0 & \rho & 0 & \rho_T v \\ \Theta w & 0 & 0 & \rho & \rho_T w \\ \Theta H - 1 & \rho u & \rho v & \rho w & \rho_T H + \rho C_p \end{pmatrix} \tag{2}$$

where $\rho_T = \frac{\partial \rho}{\partial T}\Big|_p$, $\Theta = (\frac{1}{U_r^2} - \frac{\rho_T}{\rho C_p})$, and $U_r$ is the reference velocity given by:

$$U_r = M_r a \tag{3}$$

$$M_r = \min(1, \max(kM_\infty, M)) \tag{4}$$

$a$ is the sound speed, $M_\infty$ is the free-stream Mach number, and $k$ is a constant to avoid the singularity of eigenvector matrix near stagnation points. Let $k$ be 0.5 here.

The preconditioned equations may degenerate into the traditional compressible ones for the compressible flows. For low Mach number flows, the preconditioned method relieves the disparity between the convective and acoustic wave velocities, which is beneficial for the efficiency and accuracy of the solution.

### 2.2. Temporal Discretization Method

The dual time-stepping method is introduced to solve the unsteady flow around the moving structure. The method iterates over the pseudo time step several times until it converges, then updates the flow field in the physical time step. Discretize the pseudo-temporal term $\frac{\partial \hat{W}}{\partial \tau}$ with first-order forward difference algorithm and the real-temporal term $\frac{\partial \hat{Q}}{\partial t}$ with second-order backward difference method, as a result, the governing equations can be written as:

$$\left(\frac{I}{\Delta t} + \Gamma_u^{-1m} \frac{\partial \hat{H}}{\partial \hat{W}}\right)\Delta \hat{W}^m = -\Gamma_u^{-1m}\left(\hat{R}^m + \frac{3\hat{Q}^m - 4\hat{Q}^n + \hat{Q}^{n-1}}{2\Delta t}\right) \tag{5}$$

where $\Gamma_u^m = \left(\frac{\Delta t}{\Delta \tau}\Gamma + \frac{3M}{2}\right)^m$, $M = \frac{\partial \hat{Q}}{\partial \hat{W}}$, $m$ is the iteration number of pseudo time step, and $n$ is the iteration number of the physical time step.

Implicit lower upper decomposition method, using the symmetric Gauss–Seidel method (LU-SGS) time marching scheme is employed to accelerate the convergence rate of the solution. Then update the flow field when the pseudo time-stepping iteration stops:

$$\hat{W}^{n+1} = \hat{W}^n + \Delta \hat{W} \tag{6}$$

### 3. Structural Model and Solution Method

### 3.1. Isogai Wing

The test case focuses on the cross section of a sweptback wing and investigates its aeroelastic response to the unsteady aerodynamic loads. Isogai wing section model consists in a 2D elastically mounted airfoil, which moves in pitching about its elastic axis and plunging vertically.

The structural dynamic governing equations of the elastically mounted system can be described as follows:

$$\begin{aligned} m\ddot{h} + S_\alpha \ddot{\alpha} + K_h h = -L \\ S_\alpha \ddot{h} + I_\alpha \ddot{\alpha} + K_\alpha \alpha = M \end{aligned} \tag{7}$$

where $h$ and $\alpha$ are plunging and pitching displacements, respectively, $K_h$ and $K_\alpha$ are mounted spring stiffness in two degrees of freedom, $m$ is the mass of the wing per unit span length, $S_\alpha$ and $I_\alpha$ are the static moment and the inertia moment about the elastic axis, $L$ and $M$ are the aerodynamic lift and the moment about the elastic axis, respectively. The equations can be nondimensionalized with the half-chord length b and the free-stream sound speed $\alpha_\infty$, and the resultant equations can be given as:

$$\begin{aligned} \ddot{h} + \frac{x_\alpha}{2}\ddot{\alpha} + \frac{4r_w^2 M_\infty^2}{V_f^2}h = -\frac{2}{\pi r_d}C_l \\ \ddot{h} + \frac{r_a^2}{2x_\alpha}\ddot{\alpha} + \frac{2r_a^2 M_\infty^2}{x_\alpha V_f^2}\alpha = \frac{4}{\pi r_d x_\alpha}C_m \end{aligned} \tag{8}$$

where $V_f = \frac{U_\infty}{\omega_\alpha b}$ is the reduced velocity, $r_d = m/(\rho \pi b^2)$, $r_\omega = \frac{\omega_h}{\omega_a}$, $C_l$ and $C_m$ are lift coefficient and moment coefficient respectively. Using a matrix form, Equation (8) can be written as:

$$[M]\left\{\ddot{S}\right\} + [K]\{S\} = \{F\} \tag{9}$$

where

$$[M] = \begin{bmatrix} 1 & \frac{x_\alpha}{2} \\ 1 & \frac{r_\alpha^2}{2x_\alpha} \end{bmatrix}, \ [K] = \begin{bmatrix} \frac{4r_w^2 M_\infty^2}{V_f^2} & 0 \\ 0 & \frac{2r_\alpha^2 M_\infty^2}{x_\alpha V_f^2} \end{bmatrix}, \ \{F\} = \begin{bmatrix} -\frac{2c_l}{\pi r_d} \\ \frac{4c_m}{\pi r_d x_\alpha} \end{bmatrix}, \ \{S\} = \begin{bmatrix} h \\ \alpha \end{bmatrix}$$

The nondimensional structural dynamic governing equations are solved as follows:

$$\left\{S^{p+1}\right\} = \Delta t^2 + [M]^{-1}\{F\} - \Delta t^2 [M]^{-1}[K] + 2\{S^n\} - \left\{S^{n-1}\right\} \tag{10}$$

where $p$ is subiteration step number of the pseudo time-stepping method.

### 3.2. AGARD 4445.6 Wing

Based on Rayleigh–Reiz method, the generalized structural dynamics equations can be given in the form of second-order ordinary differential equations:

$$[M]\{\ddot{q}(t)\} + [D]\{\dot{q}(t)\} + [K]\{q(t)\} = \{F(t)\} \tag{11}$$

where $\{q(t)\}$ is generalized displacement vector, $[M]$, $[D]$ and $[K]$ are generalized mass matrix, damping matrix, and stiffness matrix of the structure, respectively. The generalized force is determined as:

$$F(t) = \frac{1}{2}\rho_\infty V_\infty{}^2 \iint C_p(x, y, z, t) \Phi(x, y, z) ds \tag{12}$$

where $\Phi$ is Ritz basis function satisfying the displacement boundary conditions and having sufficient continuity, and $C_p$ is the aerodynamic load coefficient on the structure. The generalized force is a key factor coupling the structural model and the fluid model, and it depends merely on the fluid model if the linear structural model is adopted.

When the generalized displacement is solved, the actual displacement of the structure can be determined as:

$$w(x, y, z, t) = \sum_{i=1}^n q_i(t)\phi_i(x, y, z) \tag{13}$$

Taking the generalized normal mode displacement as the Ritz basis function for the generalized structural dynamic equations and Equation (11) can be transformed into linear equations:

$$\left\{\dot{S}\right\} + [A]\{S\} = \{Q\} \tag{14}$$

where

$$\{S\} = \left\{ \begin{array}{c} q(t) \\ \dot{q}(t) \end{array} \right\}, [A] = \begin{bmatrix} 0 & -1 \\ \omega^2 & 2\omega\varsigma \end{bmatrix}, \{Q\} = \left\{ \begin{array}{c} 0 \\ F(t) \end{array} \right\}$$

$\omega$ and $\varsigma$ are modal frequency and damping vector respectively. To be tightly coupled with the fluid model, the linear Equation (14) is discretized into a pseudo time-stepping scheme:

$$\left. \begin{array}{c} [B]\{\Delta S\} = -(1+\phi)\{S^p\} + (1+2\phi)\{S^n\} - \phi\{S^{n-1}\} - \Delta t\{\dot{S}^p\} \\ [B] = \begin{bmatrix} 1+\phi & -\Delta t \\ \Delta t\omega^2 & 1+\phi+2\Delta t\omega\varsigma \end{bmatrix} \\ \{\Delta S\} = \left\{S^{p+1}\right\} - \{S^p\} \end{array} \right\} \tag{15}$$

when $\phi = 0.5$ and $p \to \infty$, Equation (15) performs second-order time accuracy.

## 4. Mesh Deformation Method

For volume mesh deformation driven by surface motion in FSI problems, mesh deformation method based on radial basis functions (RBF) is a powerful solution. RBF mesh deformation method with reduced point selection proposed by Rendall can achieve high quality deformed mesh efficiently for arbitrary topology. However, the improvements in deformation efficiency cause the problems of deformation errors on deformed surfaces and inverted boundary layer meshes near the wall.

An improved option based on reduced point selection RBF method is proposed, details of the method can be found in the previous work [10]. The method selects two groups of control points from the object surfaces, and the one selected by the greedy method is used to roughly calculate the mesh position and the deformation error. Then surface points with deformation errors exceeding the tolerance are picked as the second group, which is utilized to interpolate the deformation errors to the mesh points nearby. Thus, the deformation errors on the surface are greatly reduced and the inverted boundary layer meshes near the wall are effectively avoided.

## 5. Fluid–Structure Coupling Procedure

In an aeroelasticity problem, the flexible structure vibrates in response to the unsteady aerodynamic force, and in turn, the vibration affects the flow field around the structure. By discretizing the fluid and structural dynamic governing equations with the same pseudo time-stepping method, a tightly coupled fluid structure interaction method is established to analyze the aeroelasticity problem in the time domain.

Figure 1 depicts the solution procedure of the tightly coupled FSI methodology. At the beginning of the solution, the flow field and the structure properties are initialized with given parameters. Then in each physical time step, there are several sub-iteration times between the unsteady fluid dynamic equations and structural dynamic equations. Within each physical time step, the unsteady CFD solver predicts the aerodynamic loads, according to which the structural displacements are determined. At the end of each iteration, volume mesh is updated based on the structural displacements for the next physical time step until the stop criterion is satisfied.

The surface mesh points in fluid model and structural nodes usually don't coincide at the fluid-structure interface, hence a mapping algorithm is required in the coupled method. The algorithm transfers aerodynamic loads from the aerodynamic surface to the structural nodes, and the predicted displacement on the structural nodes are interpolated to the surface mesh points of the fluid mesh. Various algorithms, including finite-plate spline (FPS) [11], infinite-plate spline (IPS) [12], multiquadric-biharmonic (MQ) [13,14], inverse isoparametric mapping (IIM) [15], and non-uniform B-spline (NUBS) [16], have been developed for exchanging information on the interface of fluid and structure. The IPS method is extensively used in programs such as MSC/NASTRAN, ENS3DAE, CFL3DAE. An RBF-based mapping algorithm [17], which is a three-dimensional extension of the IPS method, is employed for interface mapping here:

$$\left\{ S_f \right\} = [G]\{S_s\} \tag{16}$$

$$\{F_s\} = [G]^T \left\{ F_f \right\} \tag{17}$$

where the subscripts $f$ and $s$ stand for variables in fluid model and structural model, respectively, and the interpolation matrix $[G]$ is determined as follows:

$$[G] = \left[A_{fs}\right] \cdot \left[C_{ss}^{-1}\right]$$

$$[A_{fs}] = \begin{bmatrix} 1 & x_{f_1} & y_{f_1} & z_{f_1} & \phi_{f_1 s_1} & \cdots & \phi_{f_1 s_{ns}} \\ 1 & x_{f_2} & y_{f_2} & z_{f_2} & \phi_{f_2 s_1} & \cdots & \phi_{f_2 s_{ns}} \\ \vdots & \vdots & \vdots & \vdots & \vdots & \vdots & \vdots \\ 1 & x_{f_{nf}} & y_{f_{nf}} & z_{f_{nf}} & \phi_{f_{nf} s_1} & \cdots & \phi_{f_{nf} s_{ns}} \end{bmatrix}$$

$$[C_{ss}] = \begin{bmatrix} 0 & 0 & 0 & 0 & 1 & \cdots & 1 \\ 0 & 0 & 0 & 0 & x_{s_1} & \cdots & x_{s_{ns}} \\ 0 & 0 & 0 & 0 & y_{s_1} & \cdots & y_{s_{ns}} \\ 0 & 0 & 0 & 0 & z_{s_1} & \cdots & z_{s_{ns}} \\ 1 & x_{s_1} & y_{s_1} & z_{s_1} & \phi_{s_1 s_1} & \cdots & \phi_{s_1 s_{ns}} \\ \vdots & \vdots & \vdots & \vdots & \vdots & \vdots & \vdots \\ 1 & x_{s_{ns}} & y_{s_{ns}} & z_{s_{ns}} & \phi_{s_1 s_{ns}} & \cdots & \phi_{s_{ns} s_{ns}} \end{bmatrix}$$

(18)

where $\phi$ is the basis function, $nf$ is the total number of the surface mesh and $ns$ is structure nodes number.

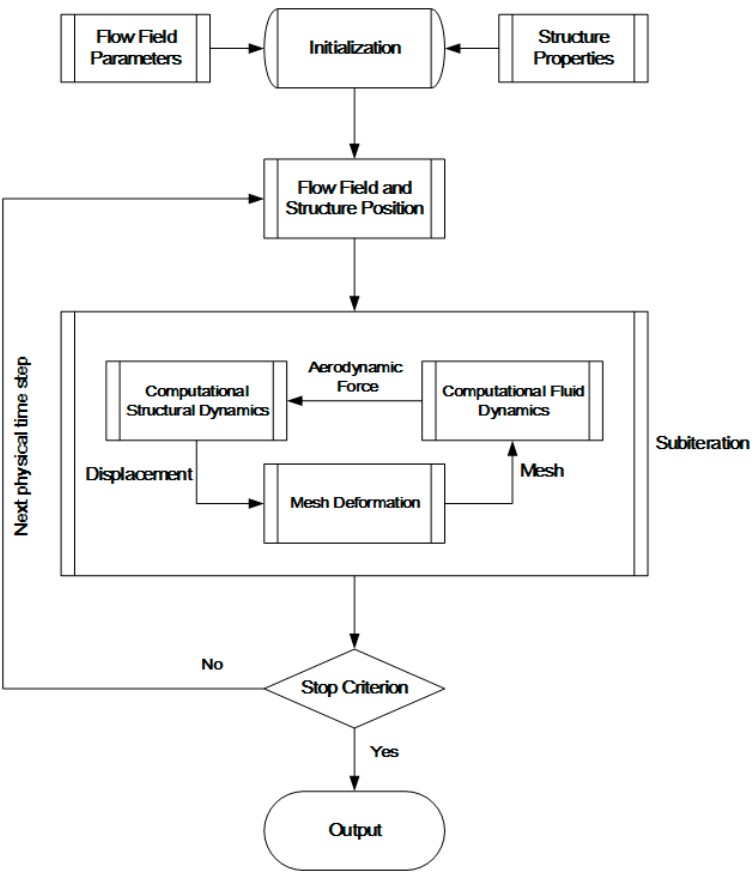

**Figure 1.** Fluid-structure coupling procedure.

## 6. Results and Discussion

In the following FSI simulations, fluid dynamics computations are performed by an in-house structured Navier–Stokes solver developed in Institute of Mechanics, Chinese Academy of Sciences for aerodynamic applications in the field of aeronautics and astronautics [6]. The flow solver, designed for both 2D and 3D CFD simulations, is a finite volume structured flow solver which solves the Reynolds-averaged Navier–Stokes equations using a cell-centered approach. Tecplot is used as data post-processing software.

To evaluate the methodology, the flutter boundary and the limit cycle oscillation of Isogai wing and the flutter boundary of AGARD 445.6 wing are analyzed.

### 6.1. Flutter Boundary and Limit Cycle Oscillation of Isogai Wing

Firstly, the tightly coupled FSI methodology is applied to analyze the Isogai wing. The model simulates the bending and torsional motion of a cross section of a sweptback wing. It consists of two DOF, plunging and pitching for a NACA 64A010 airfoil. Two aeroelastic phenomena with typically nonlinear characteristics, including the transonic flutter boundary and the LCO response, are simulated to validate the method.

The test case A of Isogai wing is investigated, and the nondimensional structural parameters are: $x_\alpha = 1.8$, $r_\omega = 1$, $r_\alpha^2 = 3.48$, $r_d = 60$ and the elastic axis located at the half-chord length. The responses to three different reduced velocities near the flutter boundary at the Mach number of 0.825 are calculated, and the corresponding displacement histories are plotted in Figure 2. The displacements in pitching and plunging coordinates oscillate with equi-amplitudes when $V^* = 0.63$, the system reaches a neutrally stable state known as flutter and the corresponding reduced velocity is the flutter speed index. At a lower reduced velocity, $V^* = 0.60$, the displacements decay with time and the system is stable for the condition. However, the displacements diverge when a higher reduced velocity is imposed, which may bring disastrous damage to the aeroelastic system. Similar simulations are implemented at different Mach numbers in transonic region, and the flutter boundary is obtained and shown in Figure 3. The speed index flutter boundary is S-shaped with a dip near the Mach number of 0.85, and the simulation result agrees well with other simulation results [18–20].

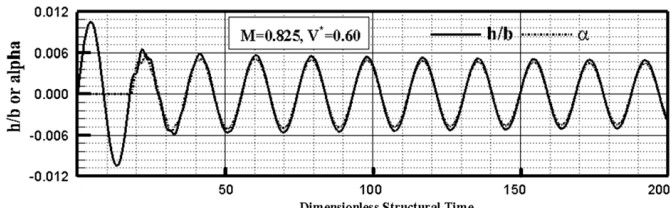

(**a**) Time history of pitching and plunging motion for V* = 0.60

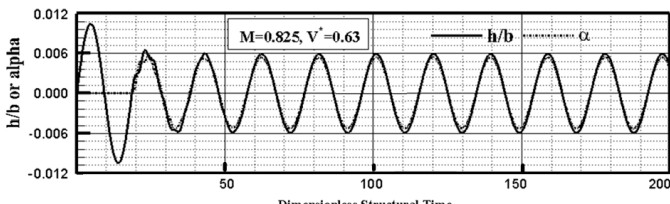

(**b**) Time history of pitching and plunging motion for V* = 0.63

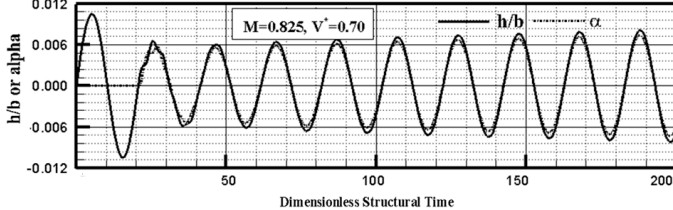

(**c**) Time history of pitching and plunging motion for V* = 0.70

**Figure 2.** Displacement responses history to different speed indexes.

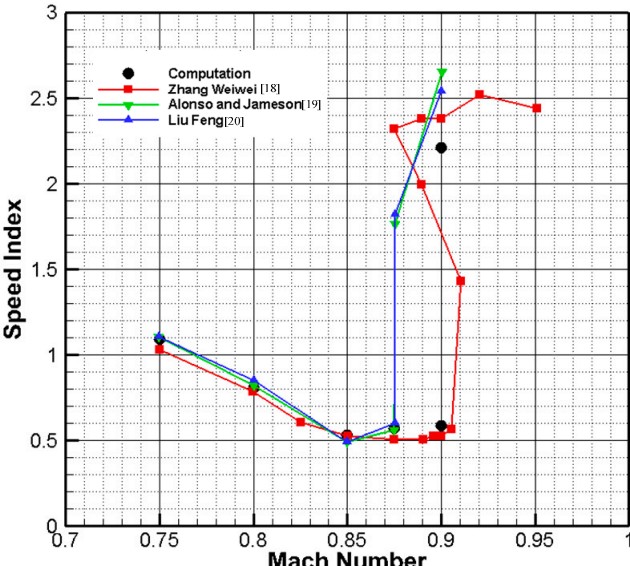

**Figure 3.** Comparison of the flutter speed index in transonic region.

For the transonic flow, the shock wave appears in the flow field around the structure, which arouses nonlinear aerodynamic loads for the structure. As a response, the aeroelastic system performs a nonlinear behavior. At the Mach number of 0.85, LCOs are observed with $V^* = 0.80$, a reduced velocity higher than the flutter speed index. During the simulation, the model is forced to oscillate sinusoidally with a given amplitude for a cycle, the plunging coordinate is activated since the second cycle and the aeroelastic response begins in the meantime. Figure 4 describes amplitudes of LCO responses to different initial movements, $\alpha_0 = 1°$, $\alpha_0 = 2°$ and $\alpha_0 = 8°$. The pitching and plunging displacements diverge in the first several cycles, then come to equi-amplitude vibrations and the amplitudes are independent of the initial movements. It is found that the responses get to the LCO state more rapidly with the initial movement amplitude increasing. Figure 5 depicts the corresponding phase portraits of the two coordinates, and the responses to different initial movements in both Dofs reach the same orbit, respectively.

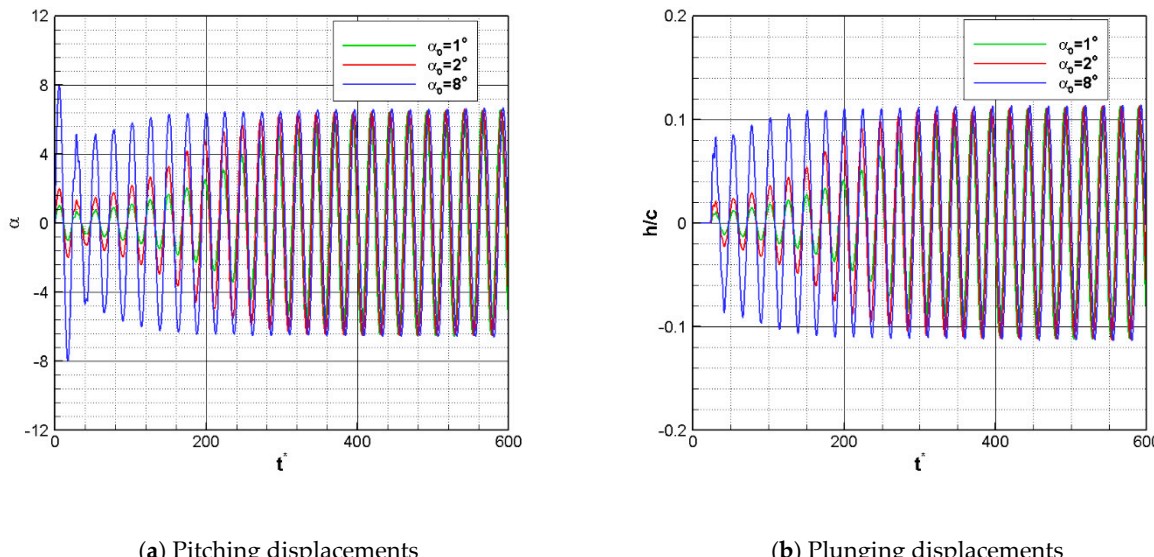

(**a**) Pitching displacements                              (**b**) Plunging displacements

**Figure 4.** LCO responses to different initial movements.

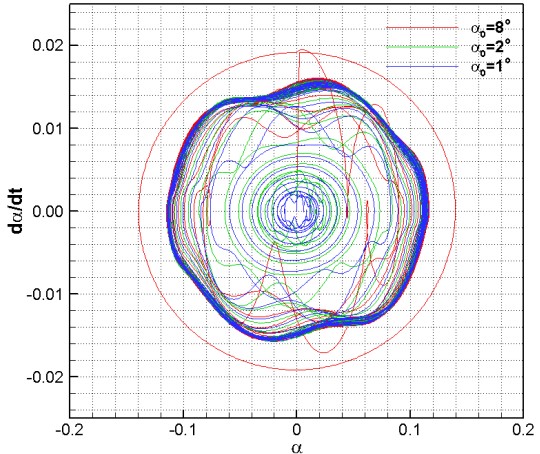
(**a**) Phase portraits of pitching displacements

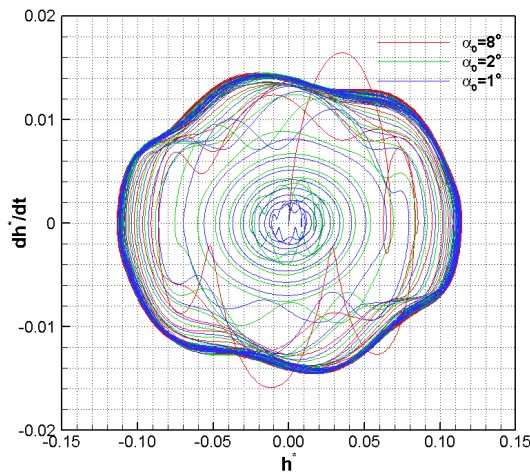
(**b**) Phase portraits of plunging displacements

**Figure 5.** Phase portraits of pitching and plunging coordinates with different initial movements.

For the two-dimensional test case, the flutter boundary in the transonic region is predicted by the tightly coupled FSI method and it is similar to those predicted by other researchers. The simulation results reveal the nonlinear response to the periodic nonlinear aerodynamic loads. The coupled CFD-CSD method can estimate the flutter boundary accurately.

### 6.2. Flutter Boundary Analysis of AGARD 445.6 Wing

AGARD 445.6 wing is a standard aeroelastic configuration and its flutter responses are extensively simulated to evaluate and assess the FSI methodology. The geometry and experimental data are provided by the NASA Langley Research Center. The flutter response of the weakened structure model mounting on the wall is analyzed.

To evaluate the uncertainties of the CFD results brought by the mesh resolution, grid independence studies are conducted on five multiblock structural meshes, named by Mesh1 with about 7000 cells, Mesh2 with about 15,030 cells, Mesh3 with about 30,000 cells, Mesh4 with about 42,000 cells and Mesh5 with 78,340 cells. In the test case, the free-stream Mach number is 0.96 and the angle of attack is $2°$. The drag coefficients corresponding to the five meshes are compared in Figure 6. It can be seen that at first the drag coefficient shows a decreasing trend with grid number increasing, then it tends towards stability even the grid number increases further. Hence, Mesh4 is employed as the final grid.

A multiblock structural fluid mesh with 42,000 cells and 121 structure nodes are adopted in the simulation, the surface mesh and the structure nodes are plotted in Figure 7. The first four vibration modes, first bending, first torsion, second bending and second torsion, are utilized in the structure dynamic model. Further, the frequencies of these four modes are 9.6 Hz, 38.17 Hz, 48.35 Hz and 91.54 Hz, respectively. The modal shapes of the four modes are shown in Figure 8.

The cross-section of the AGARD 445.6 wing is the NACA 65A004 symmetric airfoil, and the responses are simulated at the angle of attack of $0°$ with a small disturbance imposed to motivate the model at the beginning of the simulation. The dynamic aeroelastic responses at six different Mach numbers ranging from 0.499 to 1.141 are analyzed and corresponding generalized displacement responses on the flutter boundary are shown in Figure 9. On the flutter boundary of each case, all the generalized displacements perform the equi-amplitude oscillation with the same frequency, the aeroelastic system is neutral stable at the corresponding speed index.

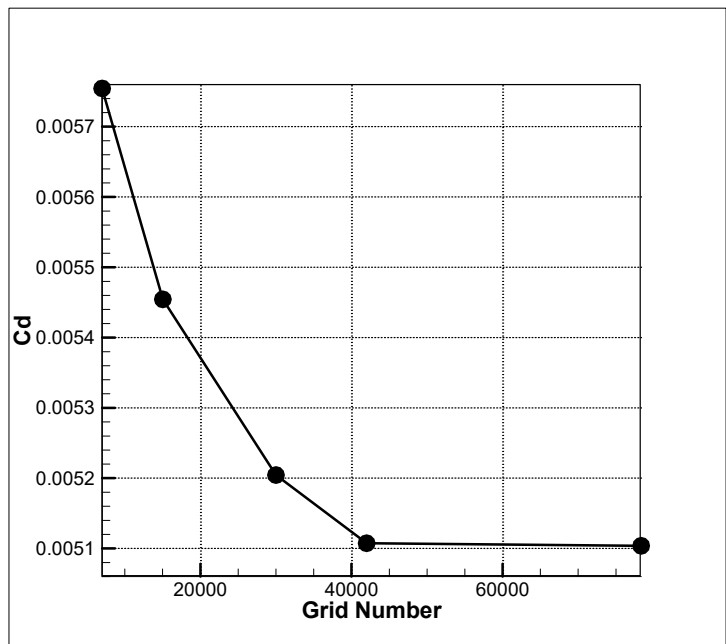

**Figure 6.** Analysis of grid independence.

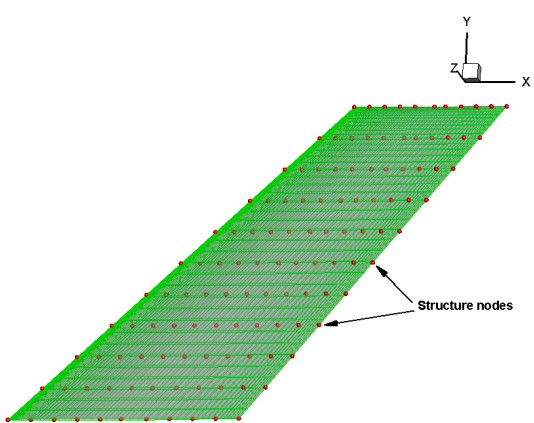

**Figure 7.** Surface mesh and structure nodes of AGARD 445.6 wing.

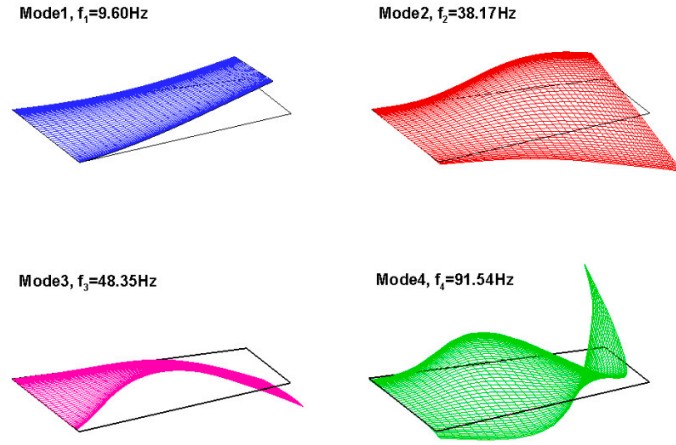

**Figure 8.** Modal shapes of the first four modes.

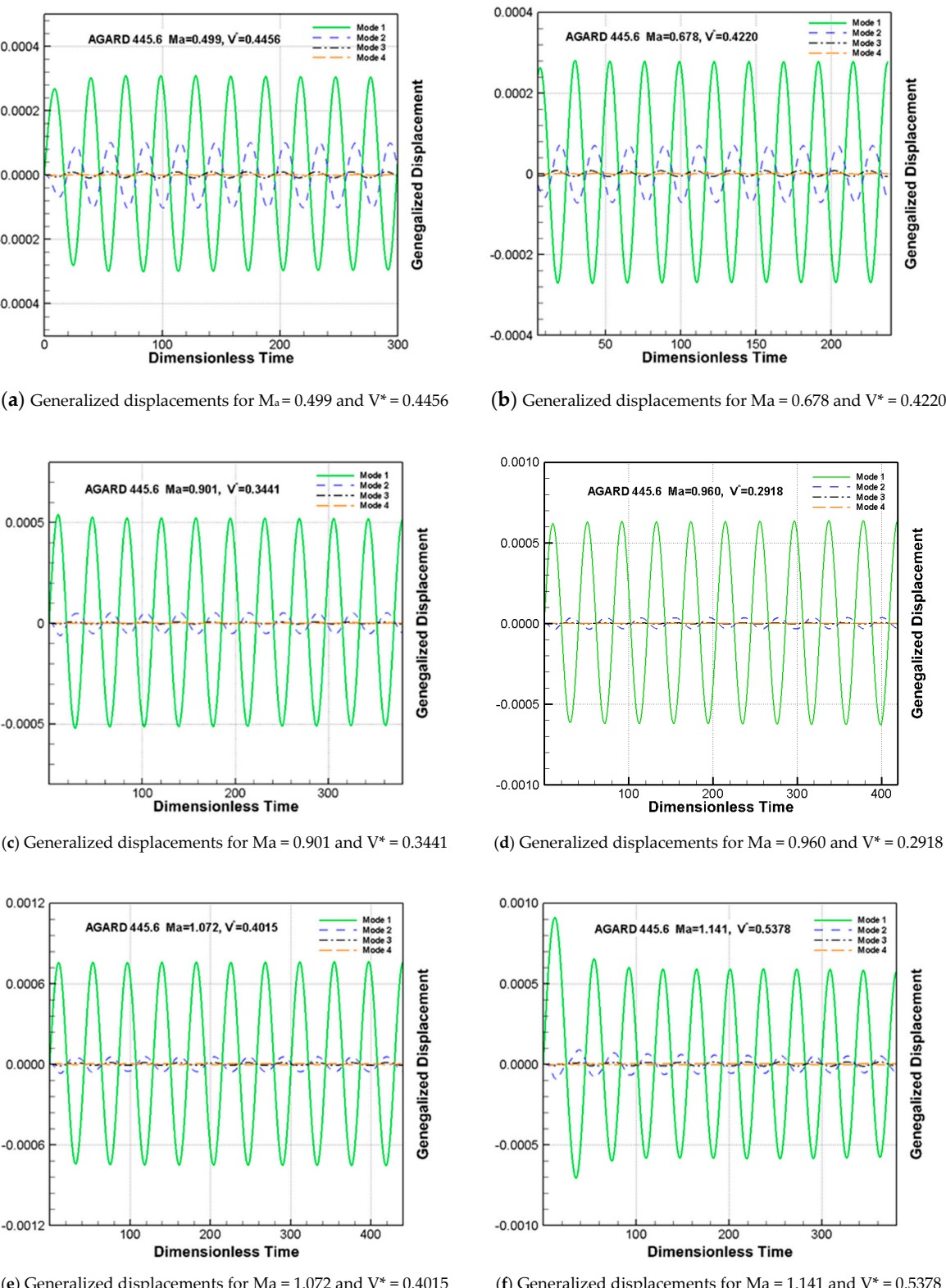

(**a**) Generalized displacements for M$_a$ = 0.499 and V* = 0.4456

(**b**) Generalized displacements for Ma = 0.678 and V* = 0.4220

(**c**) Generalized displacements for Ma = 0.901 and V* = 0.3441

(**d**) Generalized displacements for Ma = 0.960 and V* = 0.2918

(**e**) Generalized displacements for Ma = 1.072 and V* = 0.4015

(**f**) Generalized displacements for Ma = 1.141 and V* = 0.5378

**Figure 9.** Response histories of the generalized displacement on the flutter boundary.

The predicted flutter speed indexes and flutter frequency ratios are shown in Tables 1 and 2, respectively, together with the wing tunnel results. The results computed by the method proposed in the paper are in excellent agreement with the experimental data for low speed cases, M = 0.499 and M = 0.678, and the relative error of the simulation is subtle.

In transonic regime, M = 0.901 and M = 0.960, the results are reasonable with the relative error under 7%. For the supersonic cases, M = 1.072 and M = 1.141, however, coupled method overpredicts the flutter velocities by about 30% higher than the experimental data. The simulations predict the flutter frequency ratio in better accuracy compared with the flutter speed indexes, and the largest relative error is less than 10%.

**Table 1.** Comparison of predicted flutter speed index with experimental data.

| Mach Number | Computational Result | Experimental Data | Relative Error |
|---|---|---|---|
| 0.499 | 0.4456 | 0.4459 | 0.07% |
| 0.678 | 0.4220 | 0.4174 | 1.10% |
| 0.901 | 0.3441 | 0.3700 | 7.00% |
| 0.960 | 0.2918 | 0.3076 | 5.14% |
| 1.072 | 0.4015 | 0.3201 | 25.43% |
| 1.141 | 0.5378 | 0.4031 | 33.41% |

**Table 2.** Comparison of predicted frequency ratio with experimental data.

| Mach Number | Computational Result | Experimental Data | Relative Error |
|---|---|---|---|
| 0.499 | 0.5469 | 0.5353 | 2.49% |
| 0.678 | 0.4886 | 0.4722 | 3.47% |
| 0.901 | 0.4174 | 0.4224 | 1.18% |
| 0.960 | 0.3699 | 0.3648 | 1.40% |
| 1.072 | 0.3803 | 0.3623 | 4.97% |
| 1.141 | 0.4154 | 0.4592 | 9.54% |

The simulation results are also compared with those predicted by other researchers. The comparisons of flutter speed indexes and the flutter frequency ratios are shown in Figures 10 and 11. As is noted that [20,21] also overpredicted the flutter speed indexes in the supersonic region, and the larger differences are observed in those predicted results. Compared to their results, our results show some improvement by performing preconditioned Navier–Stokes calculation. The amount of improvement is relatively large, especially in the prediction of flutter frequency compared to the differences between other computational results and the experiment. The flutter frequency ratios predicted in the paper are closer to the experiment compared with those of [20,21].

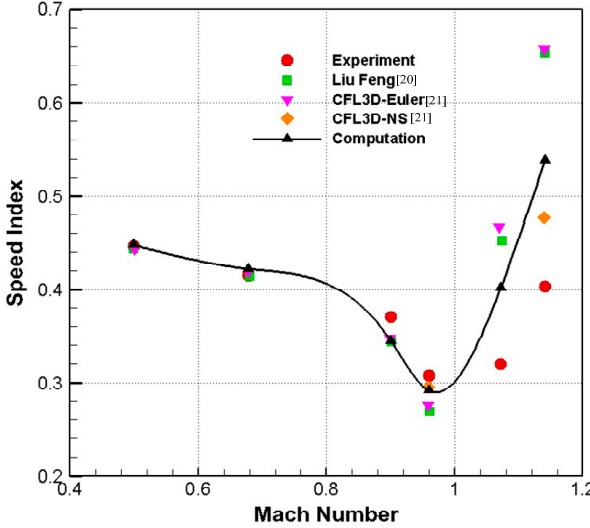

**Figure 10.** Flutter speed for the AGARD 445.6 wing.

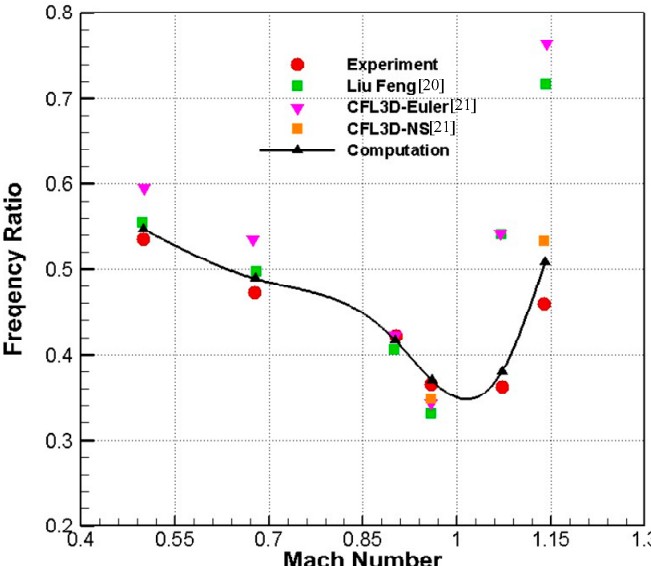

**Figure 11.** Flutter frequency for the AGARD 445.6 wing.

The results show that the tightly coupled preconditioned N-S equations with CSD methodology provides reasonable flutter speed indexes in the subsonic and the transonic regions, for the supersonic cases there is some improvement compared with other simulations. Note that the methodology predicts the flutter frequency ratios well even in the supersonic region, while other computational methods cannot still provide satisfying results in the supersonic cases.

Generally speaking, the flutter speed values are higher in the computational solutions than the experimental ones in the supersonic regime. Some researchers attribute this phenomenon to the inaccuracy of the experimental data [22,23]. It was believed that it was difficult to model correctly the physics processes occurring in the experiment at supersonic case for only minimal information on the experimental flow field was available. The differences between the predicted and measure results have also been noted in the previous studies, which is commonly believed that there exist measurement errors in the experiment of supersonic regime.

## 7. Conclusions

A tightly coupled FSI methodology based on computational fluid dynamics (CFD) and computational structural dynamics (CSD) is developed for aeroelasticity analysis in the time domain. Preconditioned N-S governing equations are used as flow solver and tightly coupled with structural model by the pseudo time-stepping temporal discretization method. A modified mesh deformation method on reduced control points RBF is utilized to update the fluid volume mesh, and an RBF based mapping algorithm is employed for data exchange on the interaction interface.

The flutter boundary of the Isogai wing Case A model is analyzed. An S-shaped flutter boundary with a dip near the Mach number of 0.85 is obtained, and the simulation results are in good agreement with those of other simulation results. The LCO response of the case is observed, with a higher reduced velocity index than the flutter speed index, the pitching and plunging displacements come to equi-amplitude oscillation finally and the vibrational amplitudes are independent of the initial movements. The responses of different initial movements in both two coordinates reach the same orbit respectively in phase portraits.

The flutter boundary of AGARD 445.6 wing is predicted. Results show that the methodology provides reasonable flutter speed indexes in the subsonic and the transonic regions, for the supersonic cases, the simulation results are overpredicted. However, some improvement is achieved by our method compared with other simulation results from

previous studies. The methodology predicts the flutter frequency ratios well even in the supersonic region as well as in subsonic and transonic regimes, while simulation results computed by other three methods still have larger differences with the experiment.

The simulation results show that the tightly coupled FSI methodology with preconditioned N-S equations and CSD models can predict the aeroelasticity in the time domain accurately, and can also reveal the nonlinear characteristics of the aeroelastic system.

**Author Contributions:** Conceptualization, Z.L., G.Y., X.N.; methodology, Z.L., X.N.; software, X.N., Z.L., G.Z.; validation, X.N., Z.L.; investigation, Z.L., X.N.; writing—original draft preparation, Z.L., X.N.; writing—review and editing, Z.L., X.N., G.Z.; supervision, G.Y., G.Z. All authors have read and agreed to the published version of the manuscript.

**Funding:** This research was funded by the National Natural Science Foundation, grant number No.11702298.

**Conflicts of Interest:** The authors declare no conflict of interest.

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
