# Peer review of "Time-Domain Aeroelasticity Analysis by a Tightly Coupled Fluid-Structure Interaction Methodology"

_applsci, doi:10.3390/app11125389_

Round 1

Reviewer 1 Report

Comment for the Authors:

The present paper presents, according to the authors, recent advances in the development of tightly coupled FSI methodology based on computational fluid dynamics (CFD)
and computational structural dynamics (CSD) for aeroelasticity analysis in
the time domain. The structural model and the fluid model are tightly coupled by the
pseudo time-stepping temporal discretization method. A modified mesh deformation
method on reduced control points RBF is utilized to update the fluid volume mesh and a
RBF based mapping algorithm is employed for data exchange on the interaction interface.

Overall this is a satisfactory paper, showing good results. I don’t have anything against publication after the following points are addressed:

  • please add reference numbers in the legends (Figures 3, 9, 10),
  • please, could you write about the new quality in your article compared to results presented in previous articles,
  • please, could you check carefully all text and figures,
  • additional small corrections are selected in the text in yellow, please check it carefully,

Reviewer 2 Report

The manuscript needs significant editing from an English style perspective (presence/absence of articles, tense shifts, etc.). A detailed listing is beyond the scope of this peer review. If this manuscript is approved, editorial staff will need to work with the authors to resolve remaining issues.

The manuscript also has many things that are grammatically incorrect (subject-verb agreement errors,  etc.) A detailed listing is beyond the scope of a peer review. The authors should be expected to resolve most of these on their own. Any remaining issues can be resolved with editorial staff if the manuscript is accepted for publication.

The authors list three different methods for coupling: fully, loosely, and tightly. However, they only define fully-coupled well. The authors should distinguish between the loosely- and tightly-coupled methods in this same paragraph.

The authors state that fully-coupled methods are impractical for 3-D problems. The citation for this statement is from 2004. If the authors believe that fully-coupled methodologies are still impractical in 2021, they should cite more recent references or their own experience.

"Mach number of infinity" should be "free-stream Mach number."

A good methodology is described adequately in the manuscript and appears to have been implemented and applied correctly on the example problems. However, there is not any advancement beyond what is already presented in the literature. A unique and significant advancement must be demonstrated for this manuscript to be worthy of archival publication in a peer-reviewed journal.

Mesh refinement studies should be included or cited. Results can only support conclusions if numerical error in the simulations is properly controlled (or at least quantified). Comparison to other researchers' results is only meaningful if differences in spatial resolution are discussed.

When results differ significantly from experimental data, the authors do not explain why. In order for this manuscript to be worthy of publication, the authors need to hypothesize why the results differ from experiment and determine what simulation parameters affect this discrepancy.

Reviewer 3 Report

The work aroused my interest as people outside the presented scope of research. I learned a lot of new things, but I cannot substantively refer to the mathematical apparatus used. I based my assessment of the article on the conclusions of the authors, which indicate that the obtained results were similar to those currently reported in the literature.

The work requires editorial correction, and some of the noticed errors are indicated in the attached file.

Reviewer 4 Report

Please use more up to date references and specify the novelty of the work in the introduction. 

The method section also needs references. Author has not invented the equations. 

What software was used to simulate the flow and for visualization?

Figures that have more than one part, need a), b) and etc. and each part should be explained in the caption and main text. 

Use standard format for tables and their borders.

Round 2

Reviewer 2 Report

Some comments from the previous review have been addressed. Some remain unadressed. Most significantly, comments from the first review regarding a lack of sufficient advancement are also applicable to the revision. There is not adequate advancement beyond what is already presented in the literature. A unique and significant advancement must be demonstrated for this manuscript to be worthy of archival publication in a peer-reviewed journal.

Reviewer 4 Report

All concerns have been addressed  

Author Response

Thank the reviewer for accepting our revision.

Round 3

Reviewer 2 Report

My original review, and the subsequent review both insisted on "major revision" before this manuscript should be reconsidered. The authors have responded with two "minor reviews." The authors need to review my comments from the first review and address them before submitting again.

Round 4

Reviewer 2 Report

See previous reviews. The manuscript has not been significantly revised.

Author Response

we feel so sorry that the revison cannot meet the requirement.